# CONVOLUTIONAL BIPARTITE ATTRACTOR NETWORKS

## ABSTRACT

In human perception and cognition, a fundamental operation that brains perform is *interpretation*: constructing coherent neural states from noisy, incomplete, and intrinsically ambiguous evidence. The problem of interpretation is well matched to an early and often overlooked architecture, the attractor network—a recurrent neural net that performs constraint satisfaction, imputation of missing features, and clean up of noisy data via energy minimization dynamics. We revisit attractor nets in light of modern deep learning methods and propose a convolutional bipartite architecture with a novel training loss, activation function, and connectivity constraints. We tackle larger problems than have been previously explored with attractor nets and demonstrate their potential for image completion and super-resolution. We argue that this architecture is better motivated than ever-deeper feedforward models and is a viable alternative to more costly sampling-based generative methods on a range of supervised and unsupervised tasks.

## 1 INTRODUCTION

Under ordinary conditions, human visual perception is quick and accurate. Studying circumstances that give rise to slow or inaccurate perception can help reveal the underlying mechanisms of visual information processing. Recent investigations of occluded (Tang et al., 2018) and empirically challenging (Kar et al., 2019) scenes have led to the conclusion that recurrent brain circuits can play a critical role in object recognition. Further, recurrence can improve the classification performance of deep nets (Tang et al., 2018; Nayebi et al., 2018), specifically for the same images with which humans and animals have the most difficulty (Kar et al., 2019).

Recurrent dynamics allow the brain to perform *pattern completion*, constructing a coherent neural state from noisy, incomplete, and intrinsically ambiguous evidence. This interpretive process is well matched to *attractor networks* (*ANs*) (Hopfield, 1982; 1984; Krotov and Hopfield, 2016; Zemel and Mozer, 2001), a class of dynamical neural networks that converge to fixed-point attractor states (Figure 1a). Given evidence in the form of a static input, an AN settles to an asymptotic state—an interpretation or completion—that is as consistent as possible with the evidence and with implicit knowledge embodied in the network connectivity. We show examples from our model in Figure 1b.

ANs have played a pivotal role in characterizing computation in the brain (Amit, 1992; McClelland and Rumelhart, 1981), not only perception (e.g., Sterzer and Kleinschmidt, 2007), but also language (Stowe et al., 2018) and awareness (Mozer, 2009). We revisit attractor nets in light of modern deep learning methods and propose a convolutional bipartite architecture for pattern completion tasks with a novel training loss, activation function, and connectivity constraints.

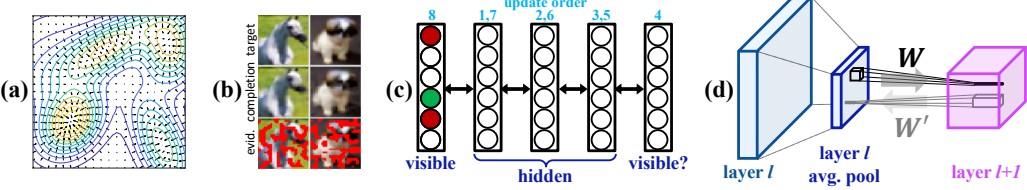

Figure 1: (a) Hypothetical activation flow dynamics of an attractor net over a 2D state space; the contours depict an energy landscape. (b) top-to-bottom: original image, completion, and evidence. (c) Bipartite architecture with layer update order. (d) Convolutional architecture with average pooling.

## 2 BACKGROUND AND RELATED RESEARCH

Although ANs have been mostly neglected in the recent literature, attractor-like dynamics can be seen in many models. For example, clustering and denoising autoencoders are used to clean up internal states and improve the robustness of deep models (Liao et al., 2016; Tang et al., 2018; Lamb et al., 2019). In a range of image-processing domains, e.g., denoising, inpainting, and super-resolution, performance gains are realized by constructing deeper and deeper architectures (e.g., Lai et al., 2018). State-of-the-art results are often obtained using deep recursive architectures that replicate layers and weights (Kim et al., 2016; Tai et al., 2017), effectively implementing an unfolded-in-time recurrent net. This approach is sensible because image processing tasks are fundamentally constraint satisfaction problems: the value of any pixel depends on the values of its neighborhood, and iterative processing is required to converge on mutually consistent activation patterns. Because ANs are specifically designed to address constraint-satisfaction problems, our goal is to re-examine them from a modern deep-learning perspective.

Interest in ANs seems to be narrow for two reasons. First, in both early (Hopfield, 1982; 1984) and recent (Li et al., 2015; Wu et al., 2018a;b; Chaudhuri and Fiete, 2017) work, ANs are characterized as content-addressable memories: activation vectors are stored and can later be retrieved with only partial information. However, memory retrieval does not well characterize the model's capabilities: like its probabilistic sibling the Boltzmann machine (Hinton, 2007; Welling et al., 2005), the AN is a general computational architecture for supervised and unsupervised learning. Second, ANs have been limited by training procedures. In Hopfield's work, ANs are trained with a simple procedure—an outer product (Hebbian) rule—which cannot accommodate hidden units and the representational capacity they provide. Recent explorations have considered stronger training procedures (e.g., Wu et al., 2018b; Liao et al., 2018); however, as for all recurrent nets, training is complicated by the issue of vanishing/exploding gradients. To facilitate training and increase the computational power of ANs, we propose a set of extensions to the architecture and training procedures.

ANs are related to several popular architectures. *Autoencoding models* such as the VAE (Kingma and Welling, 2013) and denoising autoencoders (Vincent et al., 2008) can be viewed as approximating one step of attractor dynamics, directing the input toward the training data manifold (Alain et al., 2012). These models can be applied recursively, though convergence is not guaranteed, nor is improvement in output quality over iterations. *Flow-based generative models* (FBGMs) (e.g., Dinh et al., 2016) are invertible density-estimation models that can map between observations and latent states. Whereas FBGMs require invertibility of mappings, ANs require only a weaker constraint that weights in one direction are the transpose of the weights in the other direction.

As currently formulated, *energy-based models* (EBMs) are feedforward density-estimation models that learn a mapping from input data to energies and are trained to assign low energy values to the data manifold (LeCun et al., 2006; Han et al., 2018; Xie et al., 2016; Du and Mordatch, 2019). Whereas AN dynamics are determined by an implicit energy function, the EBM dynamics are driven by optimizing or sampling from an explicit energy function. In the AN, lowering the energy for some states raises it for others, whereas the explicit EBM energy function requires well-chosen negative samples to ensure it discriminates likely from unlikely states. Although the EBM and FBGM seem well suited for synthesis and generation tasks due to their probabilistic underpinnings, we show that ANs can also be used for generation (maximum-likelihood completion) tasks.

## 3 CONVOLUTIONAL BIPARTITE ATTRACTOR NETS (CBANS)

Various types of recurrent nets have been shown to converge to activation fixed points, including fully interconnected networks of asynchronous binary units (Hopfield, 1982) and networks of continuous-valued units operating in continuous time (Hopfield, 1984). Most relevant to modern deep learning, Koiran (1994) identified convergence conditions for synchronous update of continuous-valued units in discrete time: given a network with state $x$, parallel updates of the full state with the standard activation rule,

$$x \leftarrow f(xW + b), \tag{1}$$

will asymptote at either a fixed point or a limit cycle of 2. Sufficient conditions for this result are: initial $x \in [-1, +1]^n$, $W = W^\mathsf{T}$, $w_{ii} \geq 0$, and $f(.)$ piecewise continuous and strictly increasing

with $\lim_{\eta \to \pm\infty} f(\eta) = \pm 1$. The proof is cast in terms of an energy function,

$$E(\boldsymbol{x}) = -\frac{1}{2}\boldsymbol{x}\boldsymbol{W}\boldsymbol{x}^\mathsf{T} - \boldsymbol{x}\boldsymbol{b}^\mathsf{T} + \sum_i \int_0^{x_i} f^{-1}(\xi)d\xi. \tag{2}$$

With $f \equiv \tanh$, we have the barrier function:

$$\rho(x_i) \equiv \int_0^{x_i} f^{-1}(\xi)d\xi = (1 + x_i)\ln(1 + x_i) + (1 - x_i)\ln(1 - x_i) \tag{3}$$

To ensure a fixed point (no limit cycle > 1), asynchronous updates are sufficient because the solution of $\partial E/\partial x_i = 0$ is the standard update for unit $i$ (Equation 1). Because the energy function additively factorizes for units that have no direct connections, parallel updates of these units still ensure non-increasing energy, and hence attainment of a fixed point.

We adopt the bipartite architecture of a stacked restricted Boltzmann machine (Hinton and Salakhut­dinov, 2006), with bidirectional symmetric connections between adjacent layers of units and no connectivity within a layer (Figure 1c). We distinguish between *visible* layers, which contain inputs and/or outputs of the net, and *hidden* layers. The bipartite architecture allows for units within a layer to be updated in parallel while guaranteeing strictly non-increasing energy and attainment of a local energy minimum. We thus perform layerwise updating of units, defining one *iteration* as a sweep from one end of the architecture to the other and back. The 8-step update sequence for the architecture in Figure 1c is shown above the network.

## 3.1 CONVOLUTIONAL WEIGHT CONSTRAINTS

Weight constraints required for convergence can be achieved within a convolutional architecture as well (Figure 1d). In a feedforward convolutional architecture, the connectivity from layer $l$ to $l + 1$ is represented by weights $\boldsymbol{W}^l = \{w^l_{qrab}\}$, where $q$ and $r$ are channel indices in the destination $(l + 1)$ and source $(l)$ layers, respectively, and $a$ and $b$ specify the relative coordinate within the kernel, such that the weight $w^l_{qrab}$ modulates the input to the unit in layer $l + 1$, channel $q$, absolute position $(\alpha, \beta)$—denoted $x^{l+1}_{q\alpha\beta}$—from the unit $x^l_{r,\alpha+a,\beta+b}$. If $\overline{\boldsymbol{W}}^{l+1} = \{\overline{w}^{l+1}_{qrab}\}$ denotes the reverse weights to channel $q$ in layer $l$ from channel $r$ in layer $l + 1$, symmetry requires that

$$w^l_{q,r,a,b} = \overline{w}^{l+1}_{r,q,-a,-b} \, . \tag{4}$$

This follows from the fact that the weights are translation invariant: the reverse mapping from $x^{l+1}_{q,\alpha,\beta}$ to $x^l_{r,\alpha+a,\beta+b}$ has the same weight as from $x^{l+1}_{q,\alpha-a,\beta-b}$ to $x^l_{r,\alpha,\beta}$, embodied in Equation 4. Implementation of the weight constraint is simple: $\boldsymbol{W}^l$ is unconstrained, and $\overline{\boldsymbol{W}}^{l+1}$ is obtained by transposing the first two tensor dimensions of $\boldsymbol{W}^l$ and flipping the indices of the last two. The convolutional bipartite architecture has energy function:

$$E(\boldsymbol{x}) = -\sum_{l=1}^{L-1} \sum_q \boldsymbol{x}^{l+1}_q \bullet \left(\boldsymbol{W}^l_q * \boldsymbol{x}^l\right) + \sum_{l=1}^{L} \sum_{q,\alpha,\beta} \rho(x^l_{q\alpha\beta}) - b^l_q \, x^l_{q\alpha\beta} \tag{5}$$

where $\boldsymbol{x}^l$ is the activation in layer $l$, $\boldsymbol{b}^l$ are the channel biases, and $\rho(.)$ is the barrier function (Equation 3), '$*$' is the convolution operator, and '$\bullet$' is the element-wise sum of the Hadamard product of tensors. The factor of $\frac{1}{2}$ ordinarily found in energy functions is not present in the first term because, in contrast to Equation 2, each second-order term in $\boldsymbol{x}$ appears only once. For a similar formulation in stacked restricted Boltzmann machines, see Lee et al. (2009).

## 3.2 LOSS FUNCTIONS

Evidence provided to our *Convolutional Bipartite Attractor Net* (hereafter, *CBAN*) consists of activation constraints on a subset of the visible units. The CBAN is trained to fill-in or complete the activation pattern over the visible state. The manner in which evidence constrains activations depends on the nature of the evidence. In a scenario where all features are present but potentially noisy, one should treat them as soft constraints that can be overridden by the model; in a scenario where the evidence features are reliable but other features are entirely missing, one should treat the evidence as hard constraints.

We have focused on this latter scenario in our simulations, although we discuss the use of soft constraints in Appendix A. For a hard constraint, we *clamp* the visible units to the value of the evidence, meaning that activation is set to the observed value and not allowed to change. Energy is

minimized conditioned on the clamped values. One extension to clamping is to replicate all visible units and designate one set as input, clamped to the evidence, and one set as output, which serves as the network read out. We considered using the evidence to initialize the visible state, but initialization is inadequate to anchor the visible state and it wanders. We also considered using the evidence as a fixed bias on the input to the visible state, but redundancy of the bias and top-down signals from the hidden layer can prevent the CBAN from achieving the desired activations.

An obvious loss function is squared error, $\mathcal{L}_{SE} = \sum_i ||v_i - y_i||^2$, where $i$ is an index over visible units, $\boldsymbol{v}$ is the visible state, and $\boldsymbol{y}$ is the target visible state. However, this loss misses out on a key source of error. The clamped units have zero error under this loss. Consequently, we replace $v_i$ with $\tilde{v}_i$, the value that unit $i$ would take were it unclamped, i.e., free to take on a value consistent with the hidden units driving it:

$$\mathcal{L}_{SE} = \sum_i ||\tilde{v}_i - y_i||^2.$$

An alternative loss, related to the contrastive loss of the Boltzmann machine (see Appendix B), explicitly aims to ensure that the energy of the current state is higher than that of the target state. With $\boldsymbol{x} = (\boldsymbol{y}, \boldsymbol{h})$ being the complete state with all visible units clamped at their target values and the hidden units in some configuration $\boldsymbol{h}$, and $\tilde{\boldsymbol{x}} = (\tilde{\boldsymbol{v}}, \boldsymbol{h})$ being the complete state with the visible units unclamped, one can define the loss

$$\mathcal{L}_{\Delta E} = E(\boldsymbol{x}) - E(\tilde{\boldsymbol{x}}) = \sum_i f^{-1}(\tilde{v}_i)(\tilde{v}_i - y_i) + \rho(y_i) - \rho(\tilde{v}_i).$$

We apply this loss by allowing the net to iterate for some number of steps given a partially clamped input, yielding a hidden state that is a plausible candidate to generate the target visible state. Note that $\rho(y_i)$ is constant and although it does not factor into the gradient computation, it helps interpret $\mathcal{L}_{\Delta E}$: when $\mathcal{L}_{\Delta E} = 0$, $\tilde{\boldsymbol{v}} = \boldsymbol{y}$. This loss is curious in that it is a function not just of the visible state, but, through the term $f^{-1}(\tilde{v}_i)$, it directly depends on the hidden state in the adjacent layer and the weights between these layers. A variant on $\mathcal{L}_{\Delta E}$ is based on the observation that the goal of training is only to make the two energies equal, suggesting a soft hinge loss:

$$\mathcal{L}_{\Delta E+} = \log\left(1 + \exp(E(\boldsymbol{x}) - E(\tilde{\boldsymbol{x}}))\right).$$

Both energy-based losses have an interpretation under the Boltzmann distribution: $\mathcal{L}_{\Delta E}$ is related to the conditional likelihood ratio of the clamped to unclamped visible state, and $\mathcal{L}_{\Delta E+}$ is related to the conditional probability of the clamped versus unclamped visible state:

$$\mathcal{L}_{\Delta E} = -\log\frac{p(\boldsymbol{y}|\boldsymbol{h})}{p(\tilde{\boldsymbol{v}}|\boldsymbol{h})} \qquad \text{and} \qquad \mathcal{L}_{\Delta E+} = -\log\frac{p(\boldsymbol{y}|\boldsymbol{h})}{p(\tilde{\boldsymbol{v}}|\boldsymbol{h}) + p(\boldsymbol{y}|\boldsymbol{h})}.$$

### 3.3 PREVENTING VANISHING/EXPLODING GRADIENTS

Although gradient descent is a more powerful method to train the CBAN than Hopfield's Hebb rule or the Boltzmann machine's contrastive loss, vanishing and exploding gradients are a concern as with any recurrent net (Hochreiter et al., 2001), particularly in the CBAN which may take 50 steps to fully relax. We address the gradient issue in two ways: through intermediate training signals and through a soft sigmoid activation function.

The aim of the CBAN is to produce a stable interpretation asymptotically. The appropriate way to achieve this is to apply the loss once activation converges. However, the loss can be applied prior to convergence as well, essentially training the net to achieve convergence as quickly as possible, while also introducing loss gradients deep inside the unrolled net. Assume a *stability criterion* $\theta$ that determines the iteration $t^*$ at which the net has effectively converged:

$$t^* = \min_t\left[\max_i |x_i(t) - x_i(t-1)| < \theta\right].$$

Training can be logically separated into pre- and post-convergence phases, which we will refer to as *transient* and *stationary*. In the stationary phase, the Almeida/Pineda algorithm (Pineda, 1987; Almeida, 1987) leverages the fact that activation is constant over iterations, permitting a computationally efficient gradient calculation with low memory requirements. In the transient phase, the loss can be injected at each step, which is exactly the temporal-difference method TD(1) (Sutton, 1988). Casting training as temporal-difference learning, one might consider other values of $\lambda$ in TD($\lambda$); for example, TD(0) trains the model to predict the visible state at the next time step, encouraging the model to reach the target state as quickly as feasible while not penalizing it for being unable to get to the target immediately.

Any of the losses, $\mathcal{L}_{SE}$, $\mathcal{L}_{\Delta E}$, and $\mathcal{L}_{\Delta E+}$, can be applied with a weighted mixture of training in the stationary and transient phases. Although we do not report systematic experiments in this article, we consistently find that transient training with $\lambda = 1$ is as efficient and effective as weighted mixtures

including stationary-phase-only training, and that $\lambda = 1$ outperforms any $\lambda < 1$, likely due to moving targets during training. We thus conduct all simulations with transient-phase training and $\lambda = 1$.

We propose a second method of avoiding vanishing gradients specifically due to sigmoidal activation functions: a *leaky sigmoid*, analogous to a leaky ReLU, which allows gradients to propagate through the net more freely. The leaky sigmoid has activation and barrier functions

$$f(z) = \begin{cases} \alpha(z+1) - 1 & z < -1 \\ z & -1 \le z \le 1 \\ \alpha(z-1) + 1 & z > 1 \end{cases}, \quad \rho(x) = \begin{cases} \frac{1}{2\alpha}\left[x^2 + (1-\alpha)(1+2x)\right] & \text{if } x < -1 \\ \frac{1}{2}x^2 & \text{if } -1 \le x \le 1 \\ \frac{1}{2\alpha}\left[x^2 + (1-\alpha)(1-2x)\right] & \text{if } x > 1 \end{cases}.$$

Parameter $\alpha$ specifies the slope of the piecewise linear function outside the $|x| < 1$ interval. As $\alpha \to 0$, loss gradients become flat and the CBAN fails to train well. As $\alpha \to 1$, activation magnitudes can blow up and the CBAN fails to reach a fixed point. In Appendix C, we show that convergence to a fixed point is guaranteed when $\alpha ||\boldsymbol{W}||_{1,\infty} < 1$, where $||\boldsymbol{W}||_{1,\infty} = \max_i ||\boldsymbol{w}_i||_1$. In practice, we have found that restricting $\boldsymbol{W}$ is unnecessary and $\alpha = 0.2$ works well.

## 4    SIMULATIONS

We report on a series of simulation studies of increasing complexity. First, we explore fully connected bipartite attractor net (FBAN) on a bar imputation task and then supervised MNIST image completion and classification. Second, we apply CBAN to unsupervised image completion tasks on Omniglot and CIFAR-10 and compare CBAN to CBAN-variants and denoising-VAEs. Lastly, we revision CBAN for the task of super-resolution and report promising results against competing models, such as DRCN and LapSRN. Details of architectures, parameters, and training are in Appendix D. All models are trained via SGD (back propagation through time).

### 4.1    BAR TASK

We studied a simple inference task on partial images that have exactly one correct interpretation. Images are $5 \times 5$ binary pixel arrays consisting of two horizontal bars or two vertical bars. Twenty distinct images exist, shown in the top row of Figure 2. A subset of pixels is provided as evidence; examples are shown in the bottom row of Figure 2. The task is to fill in the masked pixels. Evidence is generated such that only one consistent completion exists. In some cases, a bar must be inferred without any white pixels as evidence (e.g., second column from the right). In other cases, the local evidence is consistent with both vertical and horizontal bars (e.g., first column from left).

An FBAN with one layer of 50 hidden units is sufficient for the task. Evidence is generated randomly on each trial. Evaluating on 10k random states, the model is 99.995% correct. The middle row in Figure 2 shows the FBAN response after one iteration. The net comes close to performing the task in a single shot, but after a second iteration of clean up and the asymptotic state is shown in the top row.

Figure 3 shows some visible-hidden weights learned by the FBAN. Each $5 \times 5$ array depicts weights to/from one hidden unit. Weight sign and magnitude are indicated by coloring and area of the square, respectively. Units appear to select one row and one column, either with the same or opposite

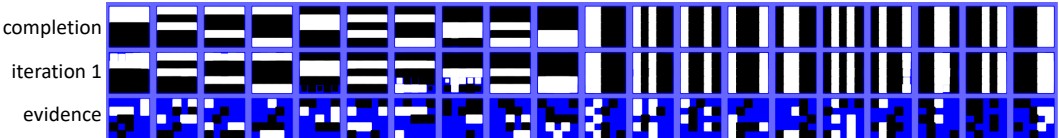

Figure 2: Bar task: Input consists of $5 \times 5$ pixel arrays with the target being either two rows or two columns of pixels present.

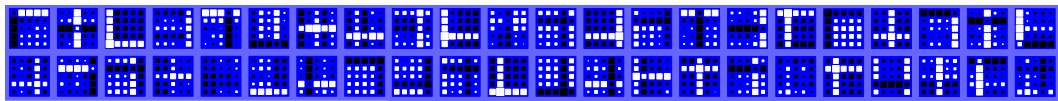

Figure 3: Bar task: Weights between visible and first hidden layers



Figure 4: MNIST completions. Row 1: target test examples, with class label coded in the bottom row. Row 2: completions produced by the FBAN. Row 3: Evidence with masked regions (including class labels) in red. Row 4: the top-down 'dream' state produced by the hidden representation.

polarity. Same-polarity weights within a row or column induce coherence among pixels. Opposite-polarity weights between a row and a column allow the pixel at the intersection to activate either the row/column depending on the sign of the unit's activation.

## 4.2 SUPERVISED MNIST

We trained an FBAN with two hidden layers on a supervised version of MNIST in which the visible state consists of a $28 \times 28$ array for an MNIST digit and an additional vector to code the class label. For the sake of graphical convenience, we allocate 28 units to the label, using the first 20 by redundantly coding the class label in pairs of units, and ignoring the final 8 units. Our architecture had 812 inputs, 200 units in the first hidden layer, and 50 units in the second. During training, all bits of the label were masked as well as one-third of image pixels. The image was masked with thresholded Perlin coherent noise (Perlin, 1985), which produces missing patches that are far more difficult to fill in than the isolated pixels produced by Bernoulli masking.

Figure 4 shows evidence provided to FBAN for 20 random test set items in the third row. The red masks indicate unobserved pixels; the other pixels are clamped in the visible state. The unobserved pixels include those representing the class label, coded in the bottom row of the pixel array. The top row of the Figure shows the target visible representation, with class labels indicated by the isolated white pixels. Even though the training loss treats all pixels as equivalent, the FBAN does learn to classify unlabeled images. On the test set, the model achieves a classification accuracy of 87.5% on Perlin-masked test images and 89.9% on noise-free test images. Note that the 20 pixels indicating class membership are no different than any other missing pixels in the input. The model learns to classify by virtue of the systematic relationship between images and labels. We can train the model with fully observed images and fully unobserved labels, and its performance is like that of any fully-connected MNIST classifier, achieving an accuracy of 98.5%.

The FBAN does an excellent job of filling in missing features in Figure 4 and in further examples in Appendix F. The FBAN's interpretations of the input seem to be respectable in comparison to other recent recurrent associative memory models (Figures 8a,b). We mean no disrespect of other research efforts—which have very different foci than ours—but merely wish to indicate we are obtaining state-of-the-art results for associative memory models. Figure 8c shows some weights between visible and hidden units. Note that the weights link image pixels with multiple digit labels. These weights stand apart from the usual hidden representations found in feedforward classification networks.

## 4.3 UNSUPERVISED OMNIGLOT

We trained a CBAN with the Omniglot data set (Lake et al., 2015) which consists of instances of 1623 characters from 50 different alphabets. The CBAN has one visible layer containing the character image, $28 \times 28 \times 1$, and three successive hidden layers with dimensions $28 \times 28 \times 128$, $14 \times 14 \times 256$, and $7 \times 7 \times 256$, all with average pooling between the layers and filters of size $3 \times 3$. Other network parameters and training details are presented in Appendix D. To vary the masking, we used random square patches of diameter 3–6, which remove on average 30% of the white pixels in the image.

We compared our CBAN to variants with critical properties removed: one without weight symmetry (*CBAN-asym*) and one in which the TD(1) training procedure is substituted for a standard squared loss at the final step (*CBAN-noTD*). We also compare to a convolutional denoising VAE (*CD-VAE*), which takes the masked image as input and outputs the completion. The CBAN with symmetric

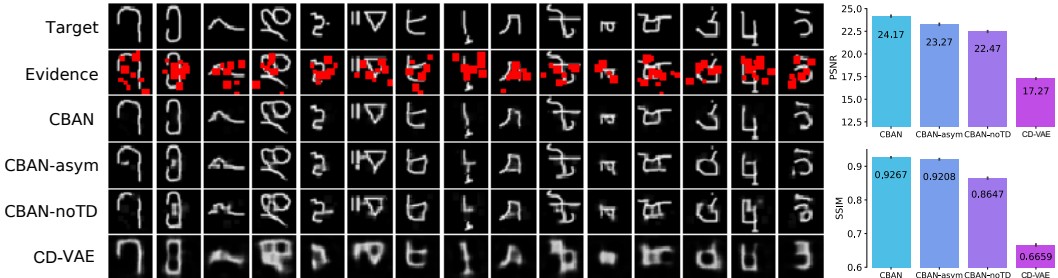

Figure 5: Omniglot image completion comparison examples (left) and quantitative results (right). The top two rows of the examples show the target image and the evidence provided to the model (with missing pixels depicted in red), respectively. The subsequent rows show the image completions produced by CBAN, CBAN-asym, CBAN-noTD, and the denoising VAE. The quantitative measures evaluate each model on PSNR and SSIM metrics; black lines indicate +1 standard error of the mean.

weights reaches a fixed point, whereas CBAN-asym appears to attain limit cycles of 2-10 iterations. Qualitatively, CBAN produces the best image reconstructions (Figure 5; additional completions in Appendix F). CBAN-asym and CBAN-noTD tend to hallucinate additional strokes; and CBAN-noTD and CD-VAE produce less crisp edges. Quantitatively, we assess models with two measures of reconstruction quality, peak signal-to-noise ratio (*PSNR*) and structural similarity (*SSIM*, Wang et al., 2004); larger is better on each measure. CBAN is strictly superior to the alternatives on both measures (Figure 5, right panel). CBAN completions are not merely memorized instances; the CBAN has learned structural regularities of the images, allowing it to fill in big gaps in images that—with the missing pixels—are typically uninterpretable by both classifiers and humans.

## 4.4 UNSUPERVISED CIFAR-10

We trained a CBAN with one visible and three hidden layers on CIFAR-10 images. The visible layer is the size of the input image, $32 \times 32 \times 3$. The successive hidden layers had dimensions $32 \times 32 \times 40$, $16 \times 16 \times 120$, and $8 \times 8 \times 440$, all with filters of size $3 \times 3$ and average pooling between the hidden layers. Further details of architecture and training can be found in Appendix D. Figure 6 shows qualitative and quantitative comparisons of alternative models. Here, CBAN-asym performs about the same as CBAN. However, CBAN-asym typically attains bi-phasic limit cycles, and CBAN-asym sometimes produces splotchy artifacts in background regions (e.g., third image from left). CBAN-noTD and the CD-VAE are clearly inferior to CBAN. Additional CBAN image completions can be found in Appendix F.

## 4.5 SUPER-RESOLUTION

Deep learning models have proliferated in many domains of image processing, perhaps none more than image super-resolution, which is concerned with recovering a high-resolution image from a

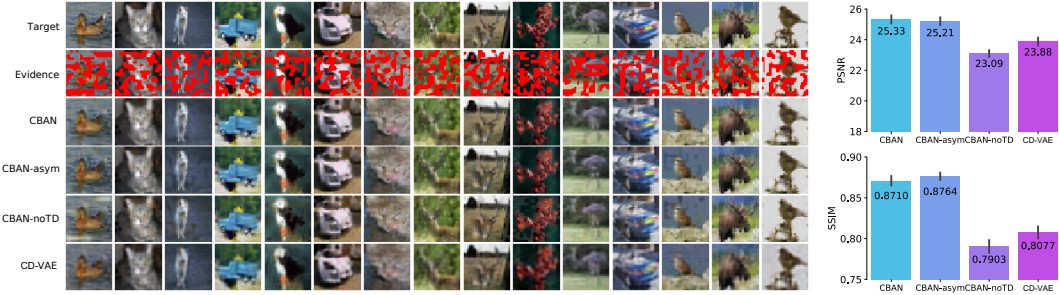

Figure 6: CIFAR-10 image completion comparison examples (left) and quantitative results (right). Layout identical to that of Figure 5.

| Algorithm | Set5 PSNR / SSIM | Set14 PSNR / SSIM | BSD100 PSNR / SSIM | Urban100 PSNR / SSIM |
|---|---|---|---|---|
| Bicubic (baseline) | 32.21 / 0.921 | 29.21 / 0.911 | 28.67 / 0.810 | 25.63 / 0.827 |
| DRCN (Kim et al., 2016) | 37.63 / 0.959 | 32.94 / 0.913 | 31.85 / 0.894 | 30.76 / 0.913 |
| LapSRN (Lai et al., 2018) | 37.52 / 0.959 | 33.08 / 0.913 | 31.80 / 0.895 | 30.41 / 0.910 |
| CBAN (ours) | 34.18 / 0.947 | 30.79 / 0.953 | 30.12 / 0.872 | 27.49 / 0.915 |

Table 1: Quantitative benchmark presenting average PSNR/SSIMs for scale factor ×2 on four test sets. Red indicates superior performance of CBAN. CBAN consistently outperforms baseline Bicubic.

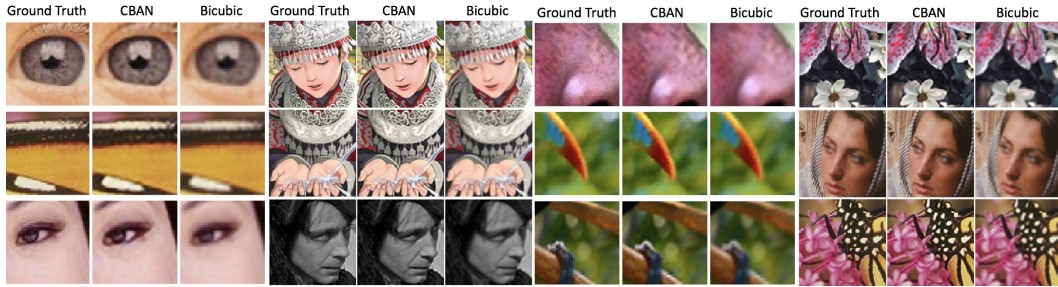

Figure 7: Examples of super-resolution, with the columns in a given image group comparing high-resolution ground truth, CBAN, and bicubic interpolation (baseline method).

low-resolution image. Many specialized architectures have been developed, and although common test data sets exist, comparisons are not as simple as one would hope due to subtle differences in methodology. (For example, even the baseline method, bicubic interpolation, yields different results depending on the implementation.) We set out to explore the feasibility of using CBANs for super-resolution. Our architecture processes $40 \times 40$ color image patches, and the visible state included both the low- and high-resolution images, with the low-resolution version clamped and the high-resolution version read out from the net. Details can be found in Appendix D.

Table 1 presents two measures of performance, SSIM and PSNR, for the CBAN and various published alternatives. CBAN beats the baseline, bicubic interpolation, on both measures, and performs well on SSIM against some leading contenders (even beating LapSRN and DRCN on *Set14* and *Urban100*), but poorly on PSNR. It is common for PSNR and SSIM to be in opposition: SSIM rewards crisp edges, PSNR rewards averaging toward the mean. The border sharpening and contrast enhancement that produce good perceptual quality and a high SSIM score (see Figure 7) are due to the fact that CBAN comes to an interpretation of the images: it imposes edges and textures in order to make the features mutually consistent. We believe that CBAN warrants further investigation for super-resolution; regardless of whether it becomes the winner in this competitive field, one can argue that it is performing a different type of computation than feedforward models like LapSRN and DRCN.

## 5 DISCUSSION

This article revisits attractor nets, which are traditionally fully interconnected RNNs trained with recurrent back propagation Liao et al. (2018) or the Hebbian rule (Hopfield, 1984). Our key contribution is to endow nets with a combination of properties which allows them to tackle difficult image completion problems; these properties include: convolutional weight constraints, novel loss functions, and methods for preventing vanishing/exploding gradients. In comparison to recent published results on image completion with attractor networks, our CBAN produces far more impressive results (see Appendix, Figure 8, for a contrast). The computational cost and challenge of training CBANs is no greater than those of training deep feedforward nets. CBANs seem to produce crisp images, on par with those produced by generative (e.g., energy- and flow-based) models. CBANs have potential to be applied in many contexts involving data interpretation, with the virtue that the computational resources they bring to bear on a task is dynamic and dependent on the difficulty of interpreting a given input. Although this article has focused on convolutional networks that have attractor dynamics *between* levels of representation, we have recently recognized the value of architectures that are fundamentally feedforward with attractor dynamics *within* a level. Our current research explores this variant of the CBAN as a biologically plausible account of intralaminar lateral inhibition.

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

APPENDIX

## A  USING EVIDENCE

The CBAN is probed with an *observation*—a constraint on the activation of a subset of visible units. For any visible unit, we must specify how an observation is used to constrain activation. The possibilities include:

- The unit is *clamped*, meaning that the unit activation is set to the observed value and is not allowed to change. Convergence is still guaranteed, and the energy is minimized conditional on the clamped value. However, clamping a unit has the disadvantage that any error signal back propagated to the unit will be lost (because changing the unit's input does not change its output).

- The unit is *initialized* to the observed value, instead of 0. This scheme has the disadvantage that activation dynamics can cause the network to wander away from the observed state. This problem occurs in practice and the consequences are so severe it is not a viable approach.

- In principle, we might try an activation rule which sets the visible unit's activation to be a convex combination of the observed value and the value that would be obtained via activation dynamics: $\alpha \times$ observed $+ (1 - \alpha) \times f$(net input). With $\alpha = 1$ this is simply the clamping scheme; with $\alpha = 0$ and appropriate start state, this is just the initialization scheme.

- The unit has an *external bias* proportional to the observation. In this scenario, the net input to a visible unit is:
$$x_i \leftarrow f(\boldsymbol{x}\boldsymbol{w}_i^\mathsf{T} + b_i + e_i), \tag{6}$$
where $e_j \propto$ observation. The initial activation can be either 0 or the observation. One concern with this scheme is that the ideal input to a unit will depend on whether or not the unit has this additional bias. For this reason the magnitude of the bias should probably be small. However, in order to have an impact, the bias must be larger.

- We might *replicate* all visible units and designate one set for input (clamped) and one set for output (unclamped). The input is clamped to the observation (which may be zero). The output is allowed to settle. The hidden layer(s) would synchronize the inputs and outputs, but it could handle noisy inputs, which isn't possible with clamping. Essentially, the input would serve as a bias, but on the hidden units, not on the inputs directly.

In practice, we have found that external biases work but are not as effective as clamping. Partial clamping with $0 < \alpha < 1$ has partial effectiveness relative to clamping. And initialization is not effective; the state wanders from the initialized values. However, the replicated-visible scheme seems very promising and should be explored further.

## B  LOSS FUNCTIONS

The training procedure for a Boltzmann machine aims to maximize the likelihood of the training data, which consist of a set of observations over the visible units. The complete states in a Boltzmann machine occur with probabilities specified by
$$p(\boldsymbol{x}) \propto e^{-E(x)/T}, \tag{7}$$
where $T$ is a computational temperature and the likelihood of a visible state is obtained by marginalizing over the hidden states. Raising the likelihood of a visible state is achieved by lowering its energy.

The Boltzmann machine learning algorithm has a contrastive loss: it tries to maximize the energy of states with the visible units clamped to training observations and minimize the energy of states with the visible units unclamped and free to take on whatever values they want. This contrastive loss is an example of an *energy-based loss*, which expresses the training objective in terms of the network energies.

In our model, we will define an energy-based loss via matched pairs of states: $\boldsymbol{x}$ is a state with the visible units clamped to observed values, and $\tilde{\boldsymbol{x}}$ is a state in which the visible units are unclamped, i.e., they are free take on values consistent with the hidden units driving them. Although $\tilde{\boldsymbol{x}}$ could be

any unclamped state, it will be most useful for training if it is related to $\boldsymbol{x}$ (i.e., it is a good point of contrast). To achieve this relationship, we propose to compute $(\tilde{\boldsymbol{x}}, \boldsymbol{x})$ pairs by:

1. Clamp some portion of the visible units with a training example.

2. Run the net to some iteration, at which point the full hidden state is $\boldsymbol{h}$. (The point of this step is to identify a hidden state that is a plausible candidate to generate the target visible state.)

3. Set $\boldsymbol{x}$ to be the complete state in which the hidden component of the state is $\boldsymbol{h}$ and the visible component is the target visible state.

4. Set $\tilde{\boldsymbol{x}}$ to be the complete state in which the hidden component of the state is $\boldsymbol{h}$ and the visible component is the fully unclamped activation pattern that would be obtained by propagating activities from the hidden units to the (unclamped) visible units.

Note that the contrastive pair at this iteration, $(\tilde{\boldsymbol{x}}_i, \boldsymbol{x}_i)$, are states close to the activation trajectory that the network is following. We might train the net only after it has reached convergence, but we've found that defining the loss for every iteration $i$ up until convergence improves training performance.

## B.1 Loss 1: The difference of energies

$$\mathcal{L}_{\Delta E} = E(\boldsymbol{x}) - E(\tilde{\boldsymbol{x}})$$

$$= \left( -\frac{1}{2}\boldsymbol{x}\boldsymbol{W}\boldsymbol{x}^\mathsf{T} - b\boldsymbol{x}^\mathsf{T} + \sum_j \int_0^{\boldsymbol{x}_j} f^{-1}(\xi)d\xi \right) - \left( -\frac{1}{2}\tilde{\boldsymbol{x}}\boldsymbol{W}\tilde{\boldsymbol{x}}^\mathsf{T} - b\tilde{\boldsymbol{x}}^\mathsf{T} + \sum_j \int_0^{\tilde{\boldsymbol{x}}_j} f^{-1}(\xi)d\xi \right)$$

$$= \sum_i (\boldsymbol{w}_i\boldsymbol{x} + b_i)(\tilde{v}_i - v_i) + \int_0^{v_i} f^{-1}(\xi)d\xi - \int_0^{\tilde{v}_i} f^{-1}(\xi)d\xi$$

$$= \sum_i f^{-1}(\tilde{v}_i)(\tilde{v}_i - v_i) + \rho(v_i) - \rho(\tilde{v}_i)$$

$$\text{with } \rho(s) = \frac{1}{2}(1 + s)\ln(1 + s) + \frac{1}{2}(1 - s)\ln(1 - s)$$

This reduction depends on $\boldsymbol{x}$ and $\tilde{\boldsymbol{x}}$ sharing the same hidden state, a bipartite architecture in which visible and hidden are interconnected, all visible-to-visible connections are zero, a tanh activation function, $f$, for all units, and symmetric weights.

The goal of this loss is to move $\tilde{\boldsymbol{x}}$ toward $\boldsymbol{x}$. Once they become identical, adjusting their relative energies is not beneficial. Thus, we propose using a hinge version of the loss:

$$\mathcal{L}_{\Delta E} = \max(0, E(\boldsymbol{x}) - E(\tilde{\boldsymbol{x}}))$$

## B.2 Loss 2: The conditional probability of correct response

This loss aims to maximize the log probability of the clamped state conditional on the choice between unclamped and clamped states. Framed as a loss, we have a negative log likelihood:

$$\mathcal{L}_{\Delta E+} = -\ln P(\boldsymbol{x} \mid \boldsymbol{x} \vee \tilde{\boldsymbol{x}})$$

$$= -\ln \frac{p(\boldsymbol{x})}{p(\tilde{\boldsymbol{x}}) + p(\boldsymbol{x})}$$

$$= \ln \left( 1 + \exp\left( \frac{E(\boldsymbol{x}) - E(\tilde{\boldsymbol{x}})}{T} \right) \right)$$

The last step is attained using the Boltzmann distribution (Equation 7).

## C  PROOF OF CONVERGENCE OF CBAN WITH LEAKY SIGMOID ACTIVATION FUNCTION

- $x = f(Wx + b)$   $x \in \mathbb{R}^n$
- $f$ is applied element wise
- $f(z) = \begin{cases} \alpha(z+1) - 1 & z < -1 \\ z & -1 \le z \le 1 \\ \alpha(z-1) + 1 & z > 1 \end{cases}$ $\in \mathbb{R}$

  where $0 < \alpha < 1$
- Assume $\|b\|_\infty \le 1$   $|b_j| \le 1$

  $\|x_t\|_\infty \le 1 + m$   where $m > 0$

  $\forall j : \|w_j\|_1 \le r$
- Then for $\|x_{t+1}\|_\infty \le m$ to hold we need:

  $\|f(Wx_t + b)\|_\infty \le 1 + m$

  $\forall j$   $|f(\underbrace{w_j^T x_t + b_j}_{z})| \le 1 + m$

  — if $|z| \le 1$ we are done

  — if $z > 1$ (analogously $z < -1$)

(1)

$f(z) = \alpha(z-1) + 1$

$\quad = \alpha(w_j^T x_t + b_j - 1) + 1$

$\quad \le \alpha(\|w_j\|_1 \|x_t\|_\infty + 1 - 1) + 1$

$\quad \le \alpha(r(1+m)) + 1$

we want the last term to be at most $1+m$:

$\alpha r(1+m) + 1 \le 1 + m$

$\alpha r \le (1 - \alpha r) m$

$\alpha r < 1 \implies \alpha < \frac{1}{r}$ → $\max_j \|w_j\|_1$

- In fact there is a degree of freedom here so we can simply use $c = \alpha r < 1$ as the only param. in the analysis. so long as we also reparameterize $b$ accordingly.

(2)

- In conclusion: given $c = \alpha r < 1$ the region of convergence must include the hypercube

  $\left[ -1 - \frac{c}{1-c}, 1 + \frac{c}{1-c} \right]$

  $\left[ -\frac{1}{1-c}, \frac{1}{1-c} \right]$

- if the barrier is set to be smaller than the above region no convergence is guaranteed

(3)

## D  NETWORK ARCHITECTURES AND HYPERPARAMETERS

### D.1  BAR TASK

Our architecture was a fully connected bipartite attractor net (FBAN) with one visible layer and two hidden layers having 48 and 24 channels. We trained using $\mathcal{L}_{\Delta E+}$ with the transient TD(1) procedure, defining network stability as the condition in which all changes in unit activation on successive iterations are less than 0.01 for a given input, tanh activation functions, batches of 20 examples (the complete data set), with masks randomly generated on each epoch subject to the constraint that only one completion is consistent with the evidence. Weights between layers $l$ and $l + 1$ and the biases in layer $l$ are initialized from a mean-zero Gaussian with standard deviation $0.1(\frac{1}{2}n_l + \frac{1}{2}n_{l+1} + 1)^{-\frac{1}{2}}$, where $n_l$ is the number of units in layer $l$. Optimization is via stochastic gradient descent with an initial learning rate of 0.01, dropped to .001; the gradients in a given layer of weights are $L_2$ renormalized to be 1.0 for a batch of examples, which we refer to as *SGD-L2*.

## D.2    MNIST

Our architecture was a fully connected bipartite attractor net (FBAN) with one visible layer to one hidden layer with 200 units to a second hidden layer with 50. We trained using $\mathcal{L}_{\Delta E+}$ with the transient TD(1) procedure, defining network stability as the condition in which all changes in unit activation on successive iterations are less than 0.01 for a given input, tanh activation functions, batches of 250 examples. Masks are generated randomly for each example on each epoch. The masks were produced by generating Perlin noise, frequency 7, thresholded such that one third of the pixels were obscured. Weights between layers $l$ and $l+1$ and the biases in layer $l$ are initialized from a mean-zero Gaussian with standard deviation $0.1(\frac{1}{2}n_l + \frac{1}{2}n_{l+1} + 1)^{-\frac{1}{2}}$, where $n_l$ is the number of units in layer $l$. Optimization is via stochastic gradient descent with learning rate 0.01; the gradients in a given layer of weights are $L_\infty$ renormalized to be 1.0 for a batch of examples, which we refer to as *SGD-Linf.* Target activations scaled to lie in [-0.999,0.999].

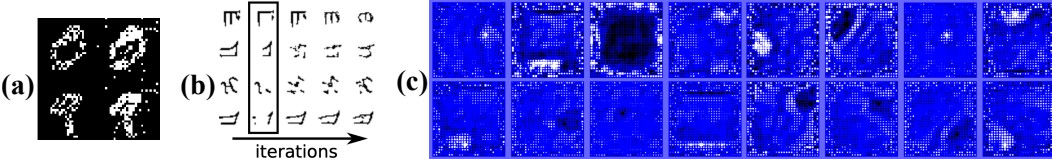

Figure 8: (a) Example of recurrent net clean-up dynamics from Liao et al. (2018). Left column is noisy input, right column is cleaned representation. (b) Example of associative memory model of Wu et al. (2018a). Column 1 is target, column 2 is input, and remaining columns are retrieval iterations. (c) Some weights between visible and first hidden layer in FBAN trained on MNIST with labels.

## D.3    OMNIGLOT

The network architecture consists of four layers: one visible layer and three hidden layers. The visible layer dimensions match the input image dimensions: $(28, 28, 1)$. The channel dimensions of the three hidden layers increase by 128, 256, and 512, respectively. We used filter sizes of $3 \times 3$ between all layers. Beyond the first hidden layer, we introduce a $2 \times 2$ average pooling operation followed by half-padded convolution going from layer $l$ to layer $l+1$, and a half-padded convolution followed by a $2 \times 2$ nearest-neighbor interpolation going from layer $l+1$ to layer $l$. Consequently, the spatial dimensions of the hidden states, from lowest to highest, are (28,28), (14,14) and (7,7). A trainable bias is applied per-channel to each layer. All biases are initialized to 0, whereas kernel weights are Gaussian initialized with a standard deviation of 0.01. The CBAN used tanh activation functions and $\mathcal{L}_{SE}$ with TD(1) transient training, as described in the main text.

We trained our model on 15,424 images from the Omniglot dataset (test set: 3856). The images are noised by online-generation of squares that mask 20-40% of the white pixels in the image. We optimized our mean-squared error objective using Adam. The learning rate is initially set to 0.0005 and then decreased manually by a factor of 10 every 20 epochs after training epoch 100. For each batch, the network runs until the state stabilizes, where the condition for stabilization is specified as the maximum absolute difference of the full network states between stabilization steps i and i+1 being less than 0.01. The maximum number of stabilization steps was set to 100; the average stabilization iteration per batch over the course of training was 50 stabilization steps.

Masks were formed by selecting patches of diameter 3–6 uniformly, in random, possibly overlapping locations, stopping when at least 25% of the white pixels have been masked.

Our DC-VAE is based on code from https://github.com/sksq96/pytorch-vae/blob/master/vae-cnn.ipynb and followed the method of Im et al. (2017).

## D.4    CIFAR-10

The network architecture consists of four layers: one visible layer and three hidden layers. The visible layer dimensions match the input image dimensions: $(32, 32, 3)$. The channel dimensions of the three hidden layers increase by 40, 120, and 440, respectively. We used filter sizes of $3 \times 3$ between all layers. Beyond the first hidden layer, we introduce a $2 \times 2$ average pooling operation followed by

half-padded convolution going from layer $l$ to layer $l + 1$, and a half-padded convolution followed by a $2 \times 2$ nearest-neighbor interpolation going from layer $l + 1$ to layer $l$. Consequently, the spatial dimensions of the hidden states, from lowest to highest, are (32,32), (16,16) and (8,8). A trainable bias is applied per-channel to each layer. All biases are initialized to 0, whereas kernel weights are Gaussian initialized with a standard deviation of 0.0001. The CBAN used tanh activation functions and $\mathcal{L}_{SE}$ with TD(1) transient training, as described in the main text.

We trained our model on 50,000 images from the CIFAR10 dataset (test set 10,000). The images are noised by online-generation of Perlin noise that masks 40% of the image. We optimized our mean-squared error objective using Adam. The learning rate is initially set to 0.0005 and then decreased manually by a factor of 10 every 20 epochs beyond training epoch 150. For each batch, the network runs until the state stabilizes, where the condition for stabilization is specified as the maximum absolute difference of the full network states between stabilization steps $t$ and $t + 1$ being less than 0.01. The maximum number of stabilization steps was set to 100; the average stabilization iteration per batch over the course of training was 50 stabilization steps.

### D.5  SUPER-RESOLUTION

The network architecture consists of four layers: one visible layer and three hidden layers. The visible layer spatial dimensions match the input patch dimensions, but consists of 6 channels: $(40, 40, 6)$. The low-resolution evidence patch is clamped to the bottom 3 channels of the visible state; the top 3 channels of the visible state serve as the unclamped output against which the high-resolution target patch is compared and loss is computed as a mean-squared error. The channel dimensions of the three hidden layers are 300, 300, and 300. We used filter sizes of $5 \times 5$ between all layers. All convolutions are half-padded and no average pooling operations are introduced in the SR network scheme. Consequently, the spatial dimensions of the hidden states remain constant and match the input patches of $(40, 40)$. A trainable bias is applied per-channel to each layer. All biases are initialized to 0, whereas kernel weights are Gaussian initialized with a standard deviation of 0.001.

We trained our model on 91 images from the T91 dataset (Yang et al., 2010) scaled at $\times 2$. We optimized our mean-squared error objective using Adam. The learning rate is initially set to 0.00005 and then decreased by a factor of 10 every 10 epochs. The stability conditions described in the CBAN models for CIFAR-10 and Omniglot are repeated for the SR task, except the stability threshold was set to 0.1 half way through training. We evaluated on four test datasets at $\times 2$ scaling: *Set5* (Bevilacqua et al., 2012), *Set14* (Zeyde et al., 2010), *DSB100* (Martin et al., 2001), and *Urban100* (Huang et al., 2015).

## E  ADDITIONAL EXPERIMENTS

### E.1  CONVERGENCE THRESHOLD

The convergence threshold operates within the forward iterations of network training and determines the point at which the network's state has satisfactorily converged. Smaller thresholds will result in longer training time, whereas large thresholds will yield faster training times. However, the higher thresholds may negatively impact network performance. We examined the effects of convergence thresholds on network performance by training two identical CBAN models on the Omniglot dataset; see § D.3 for network architecture and training details. The only difference between the models is the convergence threshold: one model uses a convergence threshold of 0.01, whereas the other uses 1.0.

Our experimental results are shown in Table 2. We find that the higher convergence threshold provides both a significant improvement in training efficiency, with an increased performance on PSNR, SSIM, and L1 reconstruction error metrics. The average number of forward iterations drops from 45 to 3 when changing from a convergence threshold of 0.01 to 1.0.

### E.2  LAYER UPDATING SCHEMES

Salakhutdinov and Hinton (2009) apply odd-even rules for layer updating in their work on Restricted Boltzmann Machines (RBMs). We empirically evaluate the odd-even update rule and compare it against a simple up-down update rule. The up-down update rule sequentially updates the layers,

| Convergence Threshold | PSNR | SSIM | L1 | Number Forward Iterations Training (Mean) | Number Forward Iterations Training (Median) |
|---|---|---|---|---|---|
| 0.01 | 24.1136 | 0.9275 | 11.7243 | 45.00 | 44.63 |
| 1.0 | **24.4494** | **0.9440** | **10.3813** | **3.01** | **3.00** |

Table 2: Quantitative comparison of variation over convergence thresholds. Higher values for PSNR and SSIM are better, whereas lower values for L1 are better. **Bold font** indicates superior performance.

starting from the bottom and moving to the top, and then back down. We explored the two alternative update rules by training two CBAN models on the Omniglot dataset; one with the up-down update rule and the other with the odd-even update rule. See § D.3 for detailed network architecture and training details. Although the odd-even rule is suggested to be a more biologically-plausible mechanism, our results show the up-down rule achieving superior performance, measured by PSNR, SSIM, and L1 reconstruction loss (see Table 3). In addition, a qualitative comparison on image completion demonstrates that the odd-even rule is less efficient when compared to the up-down rule, and more frequently fails to fully complete the task (see Figure 9).

| Update Rule | PSNR | SSIM | L1 |
|---|---|---|---|
| Odd-Even | 21.0335 | 0.9075 | 14.3044 |
| Up-Down | **22.3537** | **0.9259** | **12.0323** |

Table 3: Quantitative comparison of odd-even versus up-down update rules. Higher values for PSNR and SSIM are better, whereas lower values for L1 are better. **Bold font** indicates superior performance.

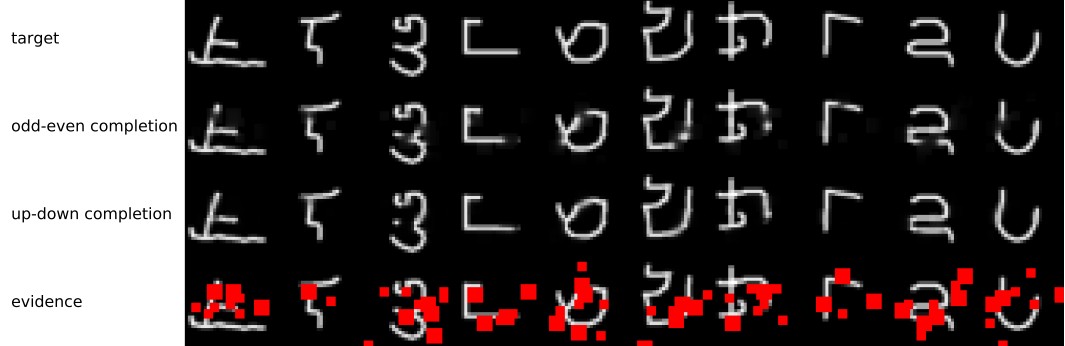

Figure 9: Update method comparisons on Omniglot. The rows of each array show the target image, the odd-even update method completion (reconstruction), the up-down update method completion (reconstruction), and the evidence provided to the CBAN with missing pixels depicted in red.

## F    ADDITIONAL RESULTS

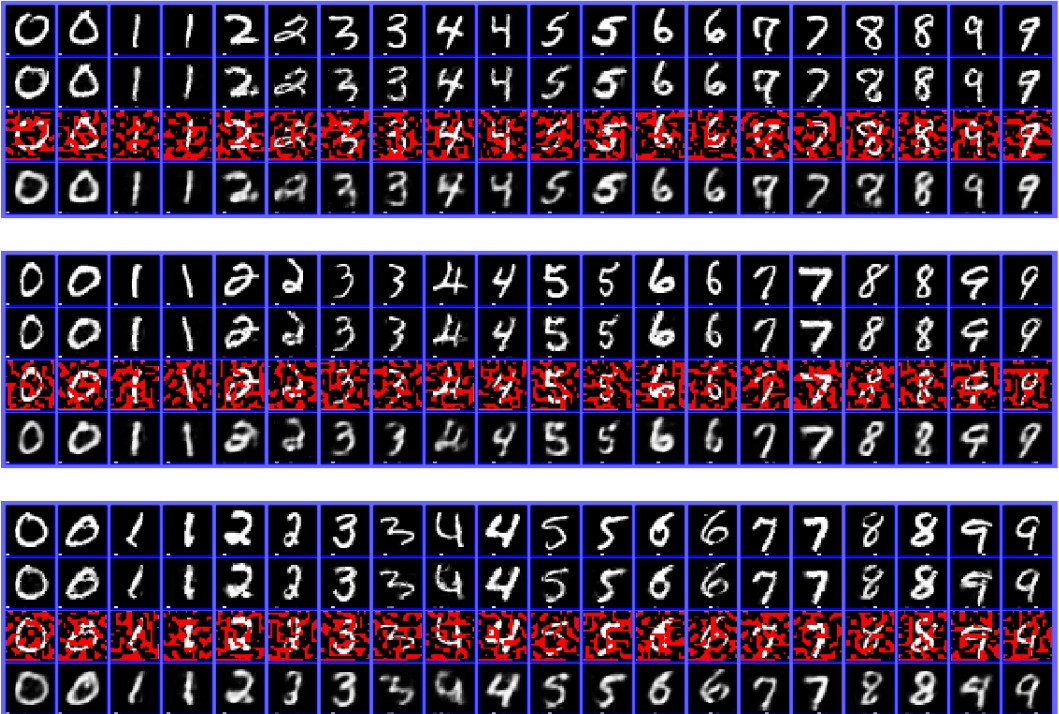

Figure 10: Additional examples of noise completion from supervised MNIST

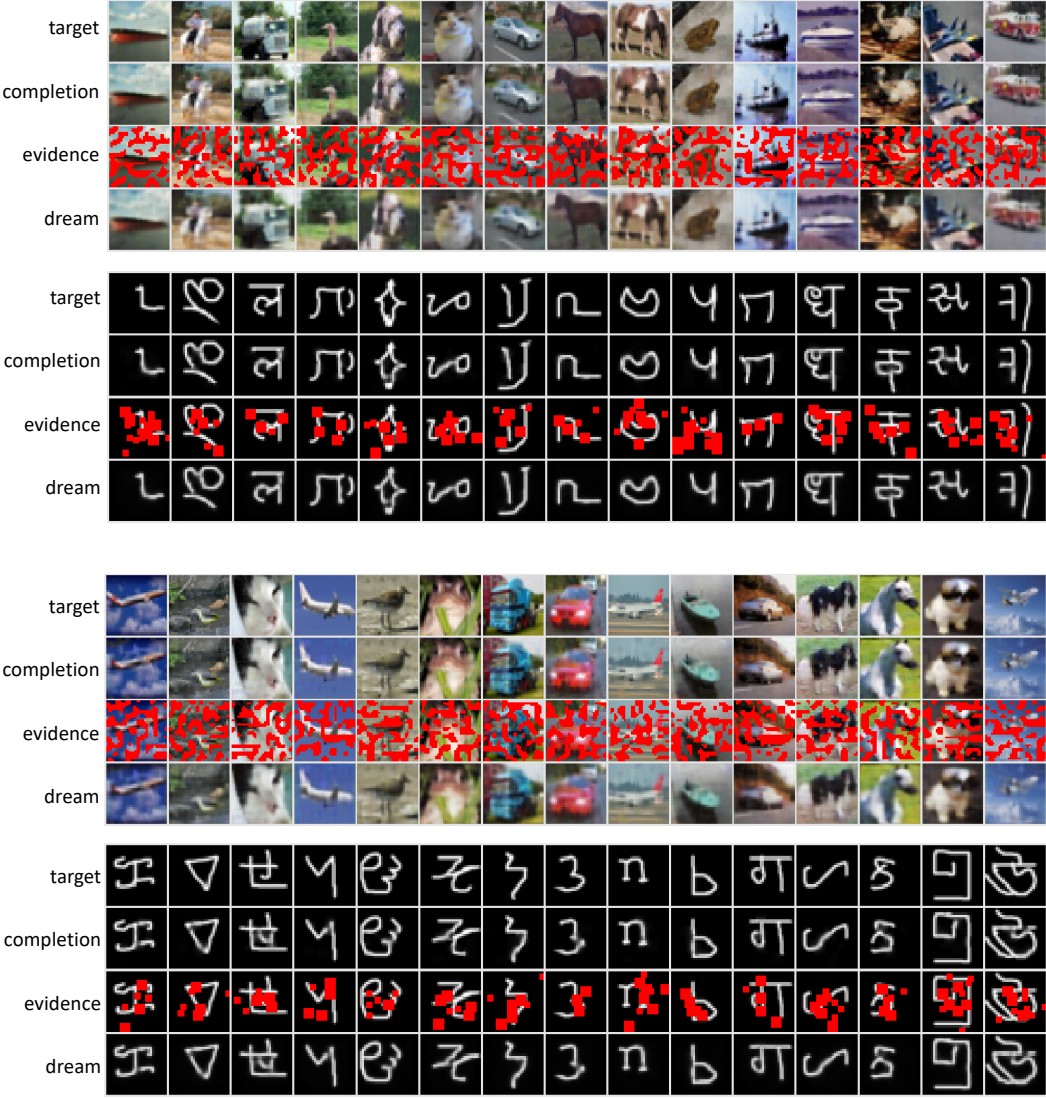

Figure 11: Additional examples of noise completion from color CIFAR-10 images and the letter-like Omniglot symbols. The rows of each array show the target image, the completion (reconstruction) produced by CBAN, the evidence provided to CBAN with missing pixels depicted in red, and the CBAN "dream" state encoded in the latent representation.

