# OpenReview forum: "Convolutional Bipartite Attractor Networks"
_ICLR.cc/2020/Conference — Reject_

### Official Review · AnonReviewer3 · 2019-10-21
**Official Blind Review #3**

**Rating:** 3

**Review:**

The paper argues for the use of attractive networks (AN) for the tasks that involve learning from noisy data. Attractor networks are recurrent in nature and use energy minimization dynamics. As motivation, the authors point to studies that give evidence for the usefulness of recurrence for visual tasks. The experiments presented show that the proposed model produces better quality images than a VAE based baseline.

The main contribution of the paper appears to the scaling of the AN based architectures to large-capacity models. The authors describe several components that are aimed at improving the gradient flow information and training stability. Firstly, a procedure to constraint the convolutional filters as required for ANs is described which allows CNNs based architectures to be used. An energy-based loss function is also described.

To tackle the vanishing/exploding gradients during training the authors propose to use leaky sigmoid activation. This is applied at the pre and post-convergence stage. How critical is the leaky sigmoid? Ablation studies for the activation choice and the loss choice need to be analyzed in more detail.

The authors should make clear the novel contributions, it can be a bit hard to extract this from the descriptions of these components. The point that the proposed model achieves state of the art results for associative memory models needs to elaborate in the text. CD-VAE seems like a weak baseline for the denoising task presented.

It will be useful to get more insights into the learning dynamics of CBAN and the ablated versions apart from the denoising performance.

Minor:

CBAN is not defined in the text.

It might be easier to follow the main body by providing all the details of a couple of tasks/datasets in the main body and moving other tasks such as super-resolution to the appendix.

Update after rebuttal: I agree with the other reviewers that the paper is not ready for publication. My score would be between 4 and 5 after the rebuttal.

The various techniques described to make the architecture work is useful.  But the experimental validation is still an issue as pointed by Reviewer 1 and 2. I do not quite agree with the authors that CD-VAE is a strong baseline.


**Experience Assessment:**

I have read many papers in this area.

**Review Assessment: Checking Correctness Of Derivations And Theory:**

I assessed the sensibility of the derivations and theory.

**Review Assessment: Checking Correctness Of Experiments:**

I assessed the sensibility of the experiments.

**Review Assessment: Thoroughness In Paper Reading:**

I made a quick assessment of this paper.

---

> ### Author Response · Authors · 2019-11-15
> **Response to Reviewer #3**
>
>
> Concerning leaky sigmoids, the bottom line is that we have not found them to be critical but their importance depends on many other decisions concerning the model. For example, if a stringent threshold for convergence is applied, we end up with (unfolded) deep networks, and pushing gradients through the network becomes challenging; in this case, the leaky sigmoids help. With a loose threshold for convergence, the resulting (unfolded) shallower networks train just fine with tanh units. This outcome also interacts with the choice of the TD(1) training procedure which injects error at each step and ensures nonzero gradients are available at every layer. The fact that each decision interacts with each other decision made it challenging for us to conduct ablation studies on one dimension at a time. We could compare configurations of decisions against one another, but one configuration is likely to be viewed as a straw person.
>
> The reviewer asks us to make clear our novel contributions. We have added a summary in the Discussion (Section 5). The clearest bird’s eye view of our contributions is section 3, with each heading summarizing a proposed architectural feature: convolutional weight constraints, novel loss functions, and methods for preventing vanishing/exploding gradients. Together, these features allow us to train more sophisticated attractor nets than have been studied before (almost always, fully connected nets trained with recurrent back prop or the Hopfield/Hebbian learning rule).
>
> The reviewer describes a CD-VAE as a weak baseline, but it is the feedforward analog of our CBAN, roughly matched in architecture to an unfolded CBAN.
>
> We agree that more ablation experiments are worth conducting (beyond those reported in Figures 5,6). We have added two sets of experiments in Appendix E that manipulate model meta-decisions, namely, the converge threshold and the layer update order.
>
> The reviewer suggested moving some experiments to the Appendix. We avoided doing so because we were concerned that reviewers would then feel there wasn’t a critical mass of experimental results.

---

### Official Review · AnonReviewer2 · 2019-10-23
**Official Blind Review #2**

**Rating:** 3

**Review:**


This paper studies and evaluates variations on attractor networks (which have recurrence and converge to a fixed point) trained end-to-end by SGD on image completion or super-resolution tasks. Several loss functions are discussed and compared.

The only quantitative results against published state-of-the-art results (on  the super-resolution task) show that the proposed approach wins on some measures and tasks and loses on others (in fact it wins 25% of the time, within a pool of 4 methods...). While the authors claim superior performance against other recent attractor networks, no quantitative comparison was found in the paper. There were quantitative comparisons against the "convolutional denoising VAE" but that model was not previously proposed by other authors to be competitive nor was it defined properly (and it is not clear from its name and the description whether this is really a VAE,  since VAEs are generative models and are not trained to denoise). Overall, I am concerned that the experimental  part of  this paper does not warrant the conclusions of superiority. That being said, I like the approach and I believe that these issues can be fixed but I'd like to see those revisions before accepting the paper.

The experiments on toy problems like the Bar Task are not necessary. Better use the place for real comparisons against published state-of-the-art benchmark baselines. Otherwise (like with your own implementation of the CD-VAE) it is plausible that the comparisons will be biased (one usually works harder on our own method than on improving someone else's architecture for our task).

Similarly, the experiments on MNIST are not very informative and you might want to push them to appendices. Also, a more interesting comparison would have been with the convolutional version, where convnets do a LOT better than 1.5% error.

The comparison against Liao et al is not fair because they don't use convolutions, which we know can make a huge difference.

In addition to the experimental results  issues, there are several places in the paper which are unclear or possibly wrong. See minor points below.

One theoretical issue which bothers me is with the "loss" L_{\Delta E}. There is no guarantee that it is lower-bounded, i.e. it can diverge which would not be appropriate for sure.

The training procedure which ends up being used is TD(1) but that procedure (in the context of the setup of the paper) has not been explained at all. It is really necessary to clarify that.

It should be clarified that the parameters are tuned by gradient descent on the loss (using backprop through time), or not?

Minor points

The last paragraph of section 2 seems strange to me. Many ANs are also EBMs so the critique of EBMs (as  if this did not apply to ANs) probably needs to be removed or reconsidered. Also the last sentence of the paragraph is clearly wrong,  since EBMs can easily handle inputs (by opposition to the outputs which are to be predicted or constructed), e.g. see conditional Boltzmann machines or the ANs in the style of Equilibrium Propagation - Scellier et al 2017 (which by the way should probably be mentioned in the review, along with follow-up work).

The 'bipartite' structure is not clear. What are the two "parts" when you have more than 2 layers? Clarify.
Note that the proposed sweep is inefficient compared to the one proposed for Deep Boltzmann Machines (Salakhutdinov et al 2009) which alternates between odd and even layers. The latter is also more biologically plausible than a feedforward-feedback sweep (consider that the differences in update times of different layers are very different depending on their depth, with your proposed scheme).

Define CBAN.

Please clarify how many iterations are needed for convergence in the various experiments. My own experience with ANs suggests that convergence can be quite slow, which is problematic both in terms of computational cost (compared to standard feedforward methods with reasonable depth) and in terms of memory. Hence the sentence in the conclusion stating that the computational cost of CBANs is no greater than DNNs is not clearly true until you demonstrate those aspects quantitatively.




**Experience Assessment:**

I have published in this field for several years.

**Review Assessment: Checking Correctness Of Derivations And Theory:**

I carefully checked the derivations and theory.

**Review Assessment: Checking Correctness Of Experiments:**

I carefully checked the experiments.

**Review Assessment: Thoroughness In Paper Reading:**

I read the paper thoroughly.

---

> ### Author Response · Authors · 2019-11-15
> **Response to Reviewer #2**
>
>
> The reviewer is concerned that our experiments do not ‘warrant the conclusions of superiority’. We agree, and we did not claim superiority. Our goal was to argue for the viability of ANs as an alternative to feedforward nets. ANs have been in existence since the early 1980s but have not received as much attention as feedforward nets; they remain outdated because of simple architectures (fully connected), training procedures (RBP at fixed points, Hebbian learning), and vanishing-gradient issues (sigmoid units). We are unaware of any attempts to update ANs and apply them to modern image processing problems. Our primary argument for the value of our work is the qualitative comparison between past applications of ANs (Figure 8) and our results (Figures 4-6), suggesting that ANs deserve a closer look. If the expectation for acceptance is that we have unequivocally outperformed non-AN state of the art methods, we acknowledge that we have not met that bar.
>
> The reviewer argues that “the comparison against Liao et al is not fair because they don’t use convolutions, which we know can make a huge difference.” We are flustered by this critique because a key contribution of our work is to introduce a convolutional architecture into the domain of attractor networks, a nontrivial feat due to the weight constraints required to ensure attractor dynamics. It sounds as if the reviewer agrees that it is a significant advance for ANs to have been formulated as multilayer convolutional networks.
>
> Concerning the model we described as the CD-VAE, our implementation is based on the work by Im, Ahn, Memisevic, and Bengio (2017, AAAI, Denoising criterion for variational autoencoder). We added this citation and the implementation source to Appendix D.3. The reviewer is concerned that we haven’t given this model a fair shot, but such a concern is true of any model comparison. The CD-VAE is the feedforward analog of our CBAN.
>
> The reviewer is concerned that the up-down sweep we use for updating CBAN state is inefficient compared to an alternating even-odd sweep (Salakhutdinov et al., 2009). We thank the reviewer for reminding us to include experiments we conducted to compare these two sweeps, which we have added to Appendix E.2. The up-down sweep yields far superior performance over the even-odd sweep for CBAN, yielding an increase in PSNR and SSIM and a decrease in L1 distance on the test set of Omniglot (Table 3), and the qualitative results are also distinguishable (Figure 9).
>
> We understand that toy problems we presented (bars, MNIST) have limited value, but they allow us to interpret weights (Figures 3, 8) to ensure networks are discovering nontrivial solutions, and solutions that appear distinct from those discovered by feedforward models.
>
> The reviewer is correct that we should be calling our model something like ‘layerwise bipartite’ or ‘k-partite’ rather than ‘bipartite’, or perhaps we  could adopt the terminology from Boltzmann machines and refer to the architecture as ‘restricted’, but “CBA Net” has a nice ring to it.
>
> We have reworded the last paragraph of Section 2 to clarify some ambiguities in our text concerning EBMs and ANs.
>
> Concerning convergence, we have added a new section, Appendix E.1, with information about convergence. Our determination of convergence is based on a threshold specifying an upper limit on the change allowed in a unit’s activation over two successive iterations. In the simulations we reported until now, the criterion was .01, and a median of 44 iterations was required to reach this criterion. However, in response to the reviewer, we also ran simulations with the much looser convergence criterion of 1.0 (for activations in  [-1,+1]). To our surprise and delight, the model not only converged quickly (median of 3 iterations), but performance greatly improved on PSNR, SSIM, and L1 distance criteria!  We are re-running all simulations with this looser convergence criterion. Why does such a loose criterion work? We suspect the reason is related to the success of Hinton’s contrastive divergence rule for Boltzmann machines, which involves taking a single up-down sweep.
>
> The reviewer is concerned about our $L_{\Delta E}$ loss which has no lower bound. We agree and in fact its semantics make more sense if we implement it as a hinge loss: $L_{\Delta E} = \max(0, E(x)-E(\tilde{x}))$, where the goal is move $\tilde{x}$ toward $x$ at which point the loss is zero. We have made a note of this proposal in Appendix B.
>
> Why does $\lambda=1$ work best for training? We did not report systematic explorations of $\lambda$ in $TD(\lambda)$, but we find monotonic improvement as \lambda varies from 0 to 1. We suspect the reason is that $\lambda < 1$ results in moving targets during training: the net is learning to predict its own future output which changes over the course of training. We have added  this hypothesis to our paper in Section 3.3.
>
> Models are indeed tuned via SGD (BPTT). We added a sentence to Section 4.

---

### Official Review · AnonReviewer1 · 2019-11-08
**Official Blind Review #1**

**Rating:** 3

**Review:**

This paper presents an attractor network (AN) approach for pattern interpretation and completion. The authors propose a convolutional bipartite architecture consisting of visible (input and output) and hidden layers with weight constraints and squared and energy-based losses. To prevent vanishing/exploding gradients, temporal-difference method and leaky sigmoid activation function are exploited. Training is done by stochastic gradient descent. In experimental validation, the proposed model is able to reconstruct missing pixels in the images for bar task and supervised MNIST. And, in OMNIGLOT and CIFAR-10 experiments, the proposed approach outperforms its variants, and in super-resolution results, it outperforms the baselines.

The approach looks interesting but the motivation of using AN might not be well justified in the current presentation. Interpretation was emphasized in the paper, but it was not clear to me what induce interpretability in the model and how to interpret experimental results. Also, even though the proposed approach shows promising results in experiments, there were no baseline or only its variants as the baseline, except super-resolution task.


Detailed comments:
Do you have any explanation why \lambda =1 in TD(\lambda) works best among others?
It was not clear to me how the proposed approach uses recurrent networks.
In supervised MNIST experiments, it was claimed that it achieved the state-of-the-art results. But figure 8 might not be enough to justify the claim and could you provide more evidence?


I am not quite familiar with this area and my understanding might be limited. But, to the best of my judgement, this paper has an interesting idea but is not yet ready to be published in ICLR.



**Experience Assessment:**

I do not know much about this area.

**Review Assessment: Checking Correctness Of Derivations And Theory:**

I assessed the sensibility of the derivations and theory.

**Review Assessment: Checking Correctness Of Experiments:**

I carefully checked the experiments.

**Review Assessment: Thoroughness In Paper Reading:**

I read the paper at least twice and used my best judgement in assessing the paper.

---

> ### Author Response · Authors · 2019-11-15
> **Response to Reviewer #1**
>
>
> Motivation for attractor nets: The introduction of our article motivates the value of recurrent dynamics for image analysis tasks. Attractor nets are a subset of recurrent networks that achieve fixed-point states. Without such fixed points, it would be unclear what to read out from the model.
>
> We’re afraid we’ve confused the reviewer with our mention of interpretation in the abstract. We are not referring to the area of ML concerned with model interpretability. To quote our definition from the first sentence of the abstract: “... interpretation: constructing coherent neural states from noisy, incomplete, and intrinsically ambiguous evidence.”
>
> The reviewer requests baselines other than variants of our model. We note that we do include a denoising autoencoder (CD-VAE) as a baseline for the Omniglot and CIFAR results. Feedforward denoising encoders are the most common architecture in use now for denoising, infilling, etc., where the input ‘noise’ is simply the corruption appropriate for the particular problem (e.g., with infilling, the corruption is missing features).
>
> Why does $\lambda=1$ work best for training? We did not report systematic explorations of $\lambda$ in $TD(\lambda)$, but we find monotonic improvement as $\lambda$ varies from 0 to 1. We suspect the reason is that $\lambda < 1$ results in moving targets during training: the net is learning to predict its own future output which changes over the course of training. We have added  this hypothesis to our paper in Section 3.3.
>
> The reviewer is not clear on why our approach uses recurrent networks. We refer the reviewer to the wikipedia page on ‘recurrent neural networks’: it is a network with temporal dynamics. Perhaps the reviewer is confounding sequence processing with recurrence. Indeed, CBAN receives a static input but has temporal dynamics.

---

### Decision · Program_Chairs · 2019-12-19

**Decision:**

Reject

**Comment:**

This paper proposes to reintroduce bipartite attractor networks and update them using ideas from modern deep net architectures.

After some discussions, all three reviewers felt that the paper did not meet the ICLR bar, in part because of an insufficiency of quantitative results, and in part because the extension was considered pretty straightforward and the results unsurprising, and hence it did not meet the novelty bar. I therefore recommend rejection.